# *Trichoderma harzianum* Cellulase Gene *thph2* Affects *Trichoderma* Root Colonization and Induces Resistance to Southern Leaf Blight in Maize

**DOI:** 10.3390/jof9121168

**Published:** 2023-12-04

**Authors:** Bo Lang, Jie Chen

**Affiliations:** 1School of Agriculture and Biology, Shanghai Jiao Tong University, Shanghai 200240, China; langbobo2@sjtu.edu.cn; 2Shanghai Yangtze River Delta Eco-Environmental Change and Management Observation and Research Station, Ministry of Science and Technology, Ministry of Education, 800 Dongchuan Rd., Shanghai 200240, China; 3The State Key Laboratory of Microbial Metabolism, Shanghai Jiao Tong University, Shanghai 200240, China

**Keywords:** *Trichoderma*, cellulase, *thph2*, colonization, ISR

## Abstract

*Trichoderma*, widely distributed all over the world, is commonly found in soil and root ecosystems. It is a group comprising beneficial fungi that improve plant disease resistance and promote plant growth. Studies have shown that *Trichoderma* cellulases can also improve plant disease resistance. Based on previous studies, we reported that a C6 zinc finger protein (Thc6) regulates two cellulase genes, *thph1* and *thph2*, to induce ISR responses in plants. Therefore, in this study, we focused on the role of *thph2* in the colonization of maize roots by *T. harzianum* and the induction of systemic resistance against southern leaf blight. The results showed that *thph2* had a positive regulatory effect on the *Trichoderma* colonization of maize roots. After the root was treated with *Trichoderma*, the leaf defense genes *AOS*, *LOX5*, *HPL*, and *OPR1* were expressed to resist the attack of *Cochliobolus heterostrophus*. The pure Thph2 protein also resulted in a similar induction activity of the *AOS*, *LOX5*, *HPL*, and *OPR1* expression in maize roots, further demonstrating that *thph2* can induce plant defense responses and that signal transduction occurs mainly through the JA signaling pathway.

## 1. Introduction

*Trichoderma* is commonly found in root ecosystems and soil. As a well-known beneficial fungus, it can promote plant growth and improve plant disease resistance [1]. In addition, *Trichoderma* interacting with a root system enables the plant’s immune system to better resist pathogens [1,2,3]. Studies have also shown that *Trichoderma* can trigger a transient increase in the number of reactive oxygen species (ROS) and the calcium levels in maize leaves, thereby activating and enhancing plant immune defenses [2,3]. As a result of this induced systemic resistance, *Trichoderma* is able to protect plants from a number of foliar pathogens in addition to soil-borne pathogens.

At present, the mechanism of the antagonistic *Trichoderma* system inducing plant leaf spot resistance mainly focuses on the identification of effect factors (activators) and an expression analysis of defense-response-related genes [4,5]. So far, more than 20 kinds of resistance-inducible effector factors (activators) have been proven to be produced by *Trichoderma*, including proteins, enzymes, indoleacetic acids, polysaccharides, chitin, secondary metabolites, etc. [6].

For example, the hydrophobic proteins Sm1/EPl1 [7] and Sm2/EPl2 produced by *T. virens* can induce the upregulated expression of the JA signal in maize leaves by acting on the roots [8,9].

Plant-cell-wall-degrading enzymes (such as endogalacturonase, cellulase, xylanase, etc.) produced by *Trichoderma* can directly affect maize roots and promote root colonization (bio-priming), the release of cell wall oligosaccharides, and the induction of resistance. Therefore, the potential role of *Trichoderma* CWDE elicitors in ISR against foliar diseases is of great interest.

Early studies have shown that the endogalacturonase (ThPG1) produced by *T. harzianum* [10] can be used as an initiator to promote *Trichoderma* root colonization and enhance leaf resistance to *Botrytis cinerea*. The leaves of tobacco, soybeans, and corn were treated with *Trichoderma* cellulase, which induced an increase in the ethylene content in the leaves.

Cellulase is composed of three main components, namely endoglucanase (EG), extranose glucanase (CBH), and β-glucosidase (BG). CBH includes two isomerases, CBHI and CBHII [11]. CBHI acts on the reducing end of cellulose, while CBHII acts on the non-reducing end of cellulose [12] to generate fibrinose. In fungi, the secretion of CBHII is not as high as that of CBHI, but its degradation efficiency for microcrystalline cellulose is twice that of CBHI [13,14]. However, there is no known effect of CBHII on inducing plant resistance.

To resist pathogen invasions, plants have evolved complex defense mechanisms to protect themselves from pathogens. These mechanisms involve recognizing pathogen-associated molecular patterns (PAMPs) and activating an immune response called PAMP-triggered immunity (PTI). At the same time, pathogens have developed effector molecules that can inhibit PTI to spread and survive in the host. Plants have evolved resistance proteins (R) as a response to this challenge. These resistance proteins can detect pathogen effectors and activate effect-triggered immunity (ETI) [15,16]. Modern plant immunology suggests that a plant’s innate immunity, or basic defense response, is caused by molecules from microbially conserved microbial-associated molecular patterns (MAMPs) or a plant immune response (MTI/DTI) stimulated by endogenous damage-associated molecular pattern (DAMP) molecules released from the damaged host plant. Host cells on the surface of the pattern recognition receptors (PRRs) often have a typical leucine-rich repeat receptor-like kinase (LRR-RK) structure of leucine-repeated sequences [17]. Recognition with MAMPs/DAMPs triggers a range of cellular and physiological responses [18,19,20,21]. Such responses include systemic acquired resistance (SAR) and induced systemic resistance (ISR), which are phenotypically similar but differ significantly at the genetic and biochemical levels. SAR is associated with the accumulation of salicylic acid (SA) [22,23], while ISR is a response to the accumulation of jasmonic acid (JA) and ethylene (ET) [24].

In previous laboratory studies, we found that a C6 zinc finger protein (Thc6) in *T. harzianum* plays a major role in the ISR of maize against *Curvularia* leaf spot. We demonstrated that two hydrolases, *thph1* and *thph2*, from *T. harzianum* are regulated by *Thc6*. However, *thph1*- and *thph2*-mediated signal transduction remain unknown. In this study, we demonstrate that *thph2* has a positive regulatory role in the colonization ability of *Trichoderma* in maize roots, that *thph2* is involved in the expression of systemic defense genes from root to leaf in maize, and that the jasmonic acid (JA)-mediated pathway induced by *Trichoderma thph2* may play a dominant role in the resistance of maize to leaf diseases. This study provides new insights into the mechanisms by which the interaction between *Trichoderma* and plant roots triggers systemic defenses against *C. heterostrophus* infections in maize.

## 2. Materials and Methods

### 2.1. Plant Growth Conditions

Maize seeds (Zhengdan958) were surface-disinfected by shaking in 75% ethanol for 15 min, followed by shaking in 2% (*w*/*v*) sodium hypochlorite (NaClO) (active ingredient) for 7 min, and were then washed five times with sterile ddH_2_O (double-distilled water). The seeds were germinated on sterilized filter papers (90 × 90 mm) placed in a plant growth incubator at 25 °C and 80% relative humidity for 60 h. 

After germination, the seedlings were grown under sterile hydroponic conditions in 530 mL tissue culture bottles containing 180 mL of sterile Hoagland’s solution with a piece of sterile support to support the seedlings [25]. Two seedlings were placed in each tissue culture bottle.

Sterile soil was used to grow the maize; 80 g of the substrate (organic seedling special substrate, Chuang De Li, China) was placed into a tissue culture bottle, 120 mL of water was added, and the seeds were planted 1 cm below the surface of the soil. Two seedlings were placed in each tissue culture bottle.

### 2.2. Fungal Strains and Culture

*T. harzianum* (T30), *Fusarium graminearum*, *C. heterostrophus*, and *Fusarium verticilloides* were provided by the Shanghai Jiao Tong University Culture Preservation Center. The fungi were stored in 25% glycerin. A 10 μL spore suspension was placed in a PDA plate. Next, the plates were placed in a 28 °C incubator for a period of 5 days to allow for activation. Following activation, the strains were then transferred onto a fresh PDA plate. The colony morphology was stabilized after being repeated 2–3 times [26,27].

### 2.3. Constructing the Vectors 

The construction methods of the gene deletion and overexpression vector are shown in Figure 1. Using a pCambia-1300 vector as the skeleton, the knockout vector was constructed by adding the sequences on both sides of *thph2* (thph2U and thph2L) and on both sides of G418 (including the Trpc promoter and Trpc terminator, with 1655 bp in total), and the target gene was knocked out by a homologous replacement. The anti-G418 mutant was obtained by screening. The overexpressed vector linked the TrpC promoter, *thph2* ORF, eGFP ORF, and Trpc terminator successively, and the anti-G418 mutant was obtained by screening. The pCambia-1300 vector was also used as the skeleton, and hygromycin B resistance (*hph*) was used as the screening marker. The mutant was verified as the correct transformant by a series of PCRs. (1) The *hph* gene (for hygromycin B resistance) and *g418* gene (for G418 resistance) were amplified with the primers Hph-F/Hph-R and G418-F/G418-R, respectively. (2) The *thph2* deletions were verified by amplifying the *thph2* genes [28].

The pCambia-1300 was inserted by the GAPDH promoter, eGFP, and Trpc terminator, followed by the NeoR (resistant to G418) or Hph. Then, we obtained pCambia-1300:eGFP:*hph* and pCambia-1300:eGFP:*g418*.

### 2.4. Transformation by ATMT and Screening of Transformants

An *A. tumefaciens*-mediated transformation (ATMT) was carried out according to the authors of [29] with some modifications. AGL1 containing a plasmid was grown in LB supplemented with kanamycin (50 μg/mL) and rifampin (20 μg/mL). After incubation at 28 °C for 24 h on a shaker (220 rpm), the bacterial cells were diluted to OD600 = 0.20, inoculated in an induction medium (IM) with MES (400 mM) and AS (200 μM), and cultured at 28 °C and 200 rpm for 6 h. Subsequently, the spores of *T. harzianum* and AGL1 cells were mixed in a ratio of 1:1, and a total of 200 μL of the mixture was spread onto a cellophane sheet and placed on an IM plate (90 mm in diameter) containing MES (400 mM) and AS (200 μM). After 48 h of incubation at 23 °C, the cellophane sheet was transferred to a CYA plate containing 300 μg/mL temetine and 200 μg/mL hygromycin B or G418 and incubated at 28 °C. Each putative mutant was then transferred to CYA media containing 300 μg/mL temetine and 200 μg/mL hygromycin B or G418 for five generations. The mitotic stability was tested by subculturing five generations on PDA media without hygromycin B [28]. After that, each mutant was transferred to PDA, the secured separation was carried out, and the obtained strains were detected and stored at minus 80 °C. 

Firstly, we transferred deletion vector 1300:Δ*thph2* and overexpression vector 1300:oe*thph2*:eGFP to T30 by ATMT, and we obtained transformants Δ*thph2 and* OE*thph2*:GFP. We transferred pCambia-1300:eGFP:*g418* and pCambia-1300:eGFP:*hph* into T30 and Δ*thph2*, respectively, and then we obtained transformants T30:GFP and Δ*thph2*:GFP.

### 2.5. Southern Blotting Analysis

The transformants were confirmed using PCR and then tested for homoplasmic status and copy number with Southern blot analysis. For Southern blot analysis, protocol of Edwin Southern was used [30]. Genomic DNA from both wildtype strains and mutant strains were extracted by CTAB and restricted with BamHI. The restricted DNA fragments were separated by gel electrophoresis on a 1% agarose gel and then transferred to a nylon membrane using a semidry transfer method. A probe of 795 bp (Gene G418) was used and was purified with a PCR purification kit. The probe was labelled and hybridized using the DIG High Prime DNA Labelling and Detection Starter kit II for chemiluminescent detection with NBT/BCIP (Roche, Mannheim, Germany), following the manufacturer’s instructions [31].

### 2.6. Determination of Trichoderma Growth and Reproductive Ability

An amount of 3 μL of the isolated strains were inoculated on PDA media; the inoculum concentration was 1 × 10^6^ cfu/mL. The PDA cultures were kept in a dark environment at a temperature of 28 °C for a period of 5 days. During this time, the diameter of the colonies and the area where spores were being produced were regularly observed and noted every 3 h. At the 5th day, a 5 mL of sterile water was used to wash off all the spores present on the medium surface. The count of *Trichoderma* spores was then determined using a hemocytometer. Each treatment was replicated three times.

### 2.7. Plate Confrontation Assay of Trichoderma and the Pathogen

Discs were removed from the edges of the *Trichoderma* and pathogen colonies using a sterile hole punch (5 mm diameter) and placed in turn on PDA plates 2.5 cm apart. All the plates were placed in an incubator at 28 °C for 5 days, and the pathogen was grown alone in PDA as a control. Then, *Trichoderma* was used to determine the inhibitory effect on pathogen growth by measuring the inhibited pathogen’s colony radius. Each counter measurement was set to 5 replicates. The inhibition rate was calculated as follows [32]: inhibition rate (%) = (colony radius of control group-colony radius of treatment group pointing to center of the plate)/colony radius of control group × 100.

### 2.8. Inoculum Preparation

T30, T30:GFP, ∆*thph2*:GFP, and OE*thph2*:GFP conidia were propagated on potato–dextrose agar (PDA) at 28 °C under a cycle of 12 h of light and 12 h of dark for 7 d to induce conidiation. The spores were collected using sterile ddH_2_O and filtered through a double layer of sterile Miracloth (Millipore Merck, Burlington, MA, USA).

### 2.9. Infection Assay

To evaluate the effect of the *T. harzianum thph2* gene on the colonization of roots by *Trichoderma*, seedlings were grown under hydroponic conditions without aeration in 530 mL tissue culture bottles containing 180 mL of sterile Hoagland’s solution with a piece of sterile support to support the seedling for 48 h after germination [33]. After seven days of maize growth, the maize roots were inoculated with a suspension of *Trichoderma* spores. Samples were taken at 0 d, 1 d, 3 d, and 5 d after inoculation. For sampling, fresh root and stem tissues were gently washed with sterile ddH_2_O, and then a 2 cm segment of the primary roots closest to the seed and a 1 cm segment of the stem were cut. Uninoculated root samples were also taken.

Conidia were added into 10% sodium CMC-Na (carboxymethyl cellulose sodium) with an adjusted concentration of 1 × 10^6^ cfu/mL. The seedlings were coated and incubated in sterile soil after germination. After 15 days of growth, the leaves were inoculated with a *C. heterostrophus* spore suspension of 5 × 10^5^ cfu/mL. For sampling, the root and stem tissues were gently washed with sterile ddH_2_O, and then 2 cm of root tissue from the geminated seed and 1 cm each of stem tissue and the inoculated leaf were cut. Uninoculated root samples were also taken.

### 2.10. Quantification of Trichoderma Root Colonization

To quantify the fungal biomass in maize roots, plant tissues were collected at the required time points, rapidly frozen in liquid N_2_, and finely ground into powder in a bead beater. Genomic DNA (gDNA) was extracted using a modified chloroform:octanol extraction protocol [34] and the fungal biomass was quantified using real-time quantitative polymerase chain reaction (RT-qPCR). The cycling conditions were 30 s at 95 °C, followed by 40 cycles at 95 °C for 10 s and 65 °C for 30 s each, 60 s at 60 °C, and 15 s at 95 °C, and the cooling step was 37 °C for 5 s. By calculating the ratio of the plant housekeeper gene *actin* (maize) to the specific gene *g418* (G418), the ∆∆C_t_ method was used to analyze the data [35]. The primers used for the qPCR analysis are shown in Appendix A.

### 2.11. Analysis of Gene Expression Using RT-qPCR

To determine the transcription levels of the fungal and plant genes in *Trichoderma* and maize, respectively, the total RNA was isolated from three bioreplicates of rapidly frozen tissues and used for an RT-qPCR analysis. Briefly, the RNA was extracted and purified using a FastPure Plant Total RNA Isolation Kit (Vazyme, Nanjing, China). The RT-qPCR was performed using the LightCycler 96 (Roche). The primers used for the RT-qPCR analysis of different plant stress marker genes are listed in Appendix A. The data were analyzed using the ∆∆C_t_ method (Livak and Schmittgen, 2001) by calculating the relative transcript levels of the stress marker genes in relation to that of the housekeeping reference gene 18Ss (maize). For the expression analysis of fungal genes, actin (*Trichoderma*) was used as the reference.

First-strand cDNA was synthesized from 1 mg of RNA using the HiScript III RT SuperMix for qPCR (+gDNA wiper) (Vazyme, Nanjing, China) according to the manufacturer’s instructions. The ChamQ Universal SYBR qPCR Master Mix (Vazyme, Nanjing, China) was used for qPCR amplification. The quantitative PCR was performed in a 96-well optical plate 7500 real-time PCR system by following the protocol in the instruction book (30 s at 95 °C, followed by 40 cycles at 95 °C for 10 s and 65 °C for 30 s each, 60 s at 60 °C, and 15 s at 95 °C, and the cooling step was 37 °C for 5 s). The data analysis was performed using three replicates of the biological samples. The error line represents the standard deviation (SD). In all cases, the measurements represent the ratio of the expression levels between each sample shown in each experiment and the control. All samples were normalized according to the housekeeper gene 18 s (maize) or actin (*Trichederma*), and all experiments were performed three times with similar results.

### 2.12. Data Analysis

All experiments were based on different replicates and were repeated at least 3 times. The graphs were built using Microsoft Office Excel and GraphPad Prism9 with standard error bars. The results show the repeated mean and the standard error of the mean.

## 3. Results

### 3.1. Construction and Identification of thph2-Deletion and -Overexpression Transformants

In early studies, we found that the T30 cellulase gene *thph2*, from a *T. harzianum* strain, was responsible for *Trichoderma*’s systemically induced plant defense responses against *Curvularina lunata*, a foliar pathogen, when plant roots were colonized by *Trichoderma* [3]. To determine the role of *thph2* in the *T. harzianum*-T30-induced systemic resistance in maize, we constructed *thph2*-deletion (KO) and -overexpression (OE) strains. We used the vector transformed by the State Key Laboratory of Microbial Metabolism, Shanghai Jiao Tong University, for gene replacement. The vector pCambia-1300:G418 was constructed to replace the entire *thph2* open reading frame (ORF) with a hygromycin B marker (Figure 1A). The OE strains transformed the TrpC promoter, *thph2* ORF, eGFP fusion protein, and TrpC terminator into *Trichoderma* using pCambia-1300:G418 (Figure 1B). We screened out some KO and OE strains using PCR gene detection and then transferred eGFP into the T30 and KO strains using pCambia-1300:eGFP.

Finally, we selected these three strains, T30:GFP, ∆*thph2*:GFP, and OE*thph2*:GFP, for subsequent experiments. Southern blotting was used to verify the copy number of the convertor, with G418 as the probe. The transformants with a single copy of the gene were selected for subsequent experiments (Figure 1C).

### 3.2. Expression of thph2 in Transformants

In order to verify whether the transformations were successful, a gene expression analysis was used to test whether the deletion or overexpression of the *thph2* gene would lead to the accumulation of transcripts resulting in the desired changes. The results showed that the expression level of *thph2* in ∆*thph2*:GFP was lower than that in the wildtype strain T30:GFP, whereas the expression level of OE*thph2*:GFP was higher than that of T30:GFP (Figure 2). The result was consistent with what we expected.

### 3.3. Evaluation of Growth, Reproduction, and Phenotype of thph2 Mutants

The mycelium growth, sporulation, and phenotype of T30, T30:GFP, ∆*thph2*:GFP, and OE*thph2*:GFP were evaluated using a plate assay. The phenotype showed no significant difference among these five strains, and the mycelium growth of ∆*thph2*:GFP was significantly lower than that of T30 (Figure 3A). After 5 days of growth, the spores were washed with the same amount of water to calculate the sporulation amount. The results showed that the number of spores of OE*thph2*:GFP, 2.2 × 10^8^ cfu/mL, was significantly less than that of other strains. In addition, the number of spores of ∆*thph2*:GFP was slightly higher than that for T30 and T30:GFP (Figure 3B). In the test of mycelium growth rate, there was no significant difference among the strains, although the mycelium growth rate of OE*thph2*:GFP was slightly faster, while that of ∆*thph2*:GFP was slightly slower (Figure 3C).

### 3.4. Evaluation of Trichoderma Strain Antagonism

In order to explore the broad-spectrum antagonism of *Trichoderma* mutants against pathogens, we selected *F. verticilloides* (causal pathogen of maize root disease), *F. graminearum* (causal pathogen of maize stem disease), and *C. heterostrophus* (causal pathogen of maize southern leaf blight) for confrontation cultures against T30:GFP, ∆*thph2*:GFP, and OE*thph*2:GFP, respectively. As with the inhibition of *F. verticilloides*, the inhibition rate of ∆*thph2*:GFP was 25.79%, which was lower than that of T30:GFP at 34.32%. The inhibition rate of OE*thph2*:GFP was 35.72%, which was higher than that of T30:GFP (Figure 4A). In addition, the inhibition of *F. graminearum* (Figure 4B) and *C. heterostrophus* (Figure 4C) showed a similar pattern, in which the antagonistic rate of OE*thph2*:GFP to the pathogens reached 66.25% and 55.57%, respectively, which was higher than T30:GFP and much higher than ∆*thph2*:GFP.

### 3.5. The Analysis of the Effects of thph2 on Trcihoderma Root Colonization and Host Defense Response Induction

In the hydroponics system, the colonization amount was detected at 1 day, 3 days, and 5 days after inoculation. The results showed that each strain did not colonize the root of maize 1 day after inoculation. At 3 days after inoculation, the colonization of each strain was greatly improved. Regarding the difference among strains in the root colonization ability, the colonization of OE*thph2*:GFP was significantly higher than that of T30:GFP and ∆*thph2*:GFP, and the colonization of ∆*thph2*:GFP was slightly lower than that of T30:GFP. At 5 days after inoculation, the colonization of each strain decreased and tended to be stable, and the colonization pattern was similar to that at 3 d after inoculation (Figure 5).

The four genes, *ZmAOS* (JA), *ZmHPL* (JA), *ZmPR1* (SA), and *ZmPR5* (SA), were selected for the evaluation of *thph2*’s contribution to the host defense response. In the early colonization period, at 1 day after inoculation (Figure 6A), all the tested hallmark defense genes were negatively expressed in the roots treated with the strains as compared to the control. At 3 days after inoculation (Figure 6B), the downregulation tendency of the defense genes was alleviated. *AOS* and *HPL* began to be positively expressed in the root, interestingly. At 5 days after inoculation (Figure 6C), *ZmPR1* and *ZmPR5* were upregulated in the root, particularly *ZmPR5*, as compared to the expression of both genes in the early days after inoculation. As to the leaf defense genes (Figure 7), most of the defense genes were upregulated to some extent as compared to the control. In particular, *ZmPR1* was the most prominently expressed, and *ZmPR5* at 5 d after inoculation was expressed to a greater extent in the OE*thph2*:GFP treatment than in the T30:GFP treatment and was also expressed more for T30:GFP than for ∆*thph2*:GFP (Figure 7C).

### 3.6. Thph2-Induced Resistance against C. heterostrophus in Maize, Mainly through ISR

In this part, we focus on the role of *thph2* in the induction of resistance in maize by a root treatment with *T. harzianum*. The effect of T30:GFP, ∆*thph2*:GFP, and OE*thph2*:GFP on the resistance of maize leaves to the maize leaf pathogen *C. heterostrophus* after infecting the *Trichoderma* roots for 2 weeks was compared. At four days after inoculation, the maize seedlings were inoculated with pathogen spores for the defense evaluation. The foliar lesions appeared earlier in the plants without *T. harzianum* or treated with ∆*thph2*:GFP, whereas the lesion size changed significantly and became smaller in plants treated with either the T30:GFP strain or OE*thph2*:GFP, which was revealed by the results shared in the detached leaf test (Figure 8A) and the living plant test (Figure 8B).

In order to elucidate the role of SA- and JA-mediated signaling pathways in T30:GFP-, ∆*thph2*:GFP-, and OE*thph2*:GFP-induced maize resistance, we detected the expression of seven well-defined defense-related genes in maize. Lipoxygenase (*LOX*) products, such as jasmonates and other oxygenated fatty acids, called oxylipins, are regarded as signals in ISR [36], and the best branches of this pathway are those started by the oxygenation of linolenic acid by 13-*LOX*s: the allene oxide synthase (*AOS*) branch and the hydroperoxide lyase (*HPL*) branch [37]. Therefore, the *LOX5 AOS* and *HPL* expression evaluations related to ISR were underlined in leaves grown from roots treated with the strains. In the infected OE*thph2*:GFP maize root, the AOS expression was significantly upregulated, while a basal expression level was detected in plants treated with T30:GFP or ∆*thph2*:GFP, and the ZmPR4 expression was similar to that of *AOS*. For *HPL*, the expression level for OE*thph2*:GFP was higher than for ∆*thph2*:GFP and T30:GFP (Figure 8C).

Meanwhile, we detected the SA-mediated signaling pathway genes ZmPR1, ZmPR5, and PAL in maize [38]. *ZmPR1* and *PAL* in the treated plants were expressed at similar levels to the untreated (control) plants. For *ZmPR5*, the expression level of each treatment was increased, but there was no significant difference from the control group (Figure 8D).

### 3.7. Thph2 Protein Induces Defense Response Gene Expression in Maize

To further determine the function of *thph2*, we purified the pure Thph2 protein with a concentration of 10 μg/mL and inoculated each plant with 100 μL. The roots and leaves were sampled every 12 h after the inoculation. The results showed that the expression levels of *AOS*, *HPL*, *LOX5*, and *OPR1* were significantly increased at 12 h after the inoculation, and the expression levels were the highest at 12 h (Figure 9A). The SA pathway marker genes *ZmPR1*, *ZmPR5*, and *PAL* were expressed very little (Figure 9B). Furthermore, it was indicated that the expression levels of *AOS* and *LOX5* in the leaves increased but were lower than those in the roots. *ZmPR1*, *ZmPR5*, and *PAL* in the leaves were expressed to varying degrees. As a result, it was concluded that the Thph2 protein mainly induces high expression levels of the JA pathway genes *AOS* and *HPL* in the roots, which is similar to the results obtained for the *thph2*-overexpressed strains. Therefore, this confirms the role of *thph2* in the ISR of maize plants against pathogen infections.

## 4. Discussion

*Trichoderma* spp. are widely distributed fungi and are commonly used as biocontrols against a range of plant diseases affecting corn [39], wheat [40], cucumbers, etc. [41]. At the same time, *Trichoderma* can produce resistance by secreting inductance effectors and interacting with plant roots. Based on previous studies, we previously reported that a C6 zinc finger protein (*Thc6*) regulates two hydrolase genes, *thph1* and *thph2*, to induce ISR responses in maize from the roots to the leaves. In this study, we constructed the *T. harzianum* mutants T30:GFP, ∆*thph2*:GFP, and OE*thph2*:GFP, representing the wildtype, *thph2*-deletion, and *thph2*-overexpression strains, respectively.

By detecting the sporulation level of the *thph2* mutants, it was determined that the OE*thph2*:GFP mutant yielded significantly fewer spores than T30:GFP, and the spore number yielded by ∆*thph2*:GFP was slightly higher than that of T30:GFP. It may be related to random integration with the OE*thph2*:GFP vector. In the plate confrontation assays of *Trichoderma* with *F. graminearum*, *C. heterostrophus*, and *F. verticilloides*, it was found that the inhibition rate of OE*thph2*:GFP was higher than that of T30:GFP and significantly higher than that of ∆*thph2*:GFP against the different tested pathogens.

A high expression of celluase may benefit the cell-wall-cellulose degradation of some pathogens. Even though there are fewer cellulose components in fungal pathogens, the comparatively high cellulase gene expression of *Trichoderma* would be more significant for breaking down oomycete cell walls, where cellulose is the dominant component. However, it may by possible that *Trichoderma thph2* indirectly improves the antagonistic substance production by obtaining more glucose from the medium. Since fungal-cell-wall-degrading enzymes include glucanase [42], chitinase [43], mannanase [44,45], etc., it was speculated that the overexpression of *thph2* may increase the activities of several CWDEs, such as chitinase and glucanase, in some unknown ways, which can promote the degradation of the cell wall of pathogens and inhibit their growth.

In previous studies, we observed that deletion of *thph1* and *thph2* reduced the colonization of *Trichoderma* by confocal observations [3], which was consistent with qPCR results (Figure 4C). The secretion of Thph1 cellulase in maize root was observed by immunogold electron microscopy. No detection was found in maize root treated with Δ*thph1* [3]. We speculate that this is the main reason for the decrease of *Trichoderma* colonized in maize roots. In our study, we found an interesting phenomenon where, in the interaction between *Trcihoderma* and the host roots, the root defense or immunity gene was first downregulated and then upregulated; interestingly, the *thph2* gene was closely involved in the shift in the down- and upregulation of the defense genes in the roots. This phenomenon has already been found in other authors’ work [46]. It was implied that overexpressed *thph2* in *Trichoderma* can somehow inhibit the host plant’s immune system at an early stage of the *Trichoderma*–host plant interaction, which would allow *Trichoderma* to colonize the roots more easily. Once *Trichoderma* successfully colonized the root, the *thph2* gene would shift its function to support *Trichoderma*-induced host defense gene expression against pathogen infections. However, it is still unknown in what way *thph2* is involved in the function shift.

In previous studies, we detected the lesion areas in maize leaves that were infected with *Curvularia lunata,* treated with WT and mutant strains in the early stage. The results showed that the area of leaf lesions increased when treated with single or double mutations of Δ*thph1* or Δ*thph2*. However, the area of lesions decreased when treated with the WT strain [3]. In this study, similar results were obtained by infecting maize leaf with *C. heterostrophus* (Figure 8A,B). These results suggest that these genes have a significant impact on the disease resistance of maize leaves.

A group of researchers demonstrated that *Trichoderma*-induced resistance mainly depends on the JA/ET pathway. *Opr7*, *ZmPr4*, *Aoc1*, and *Erf1* are confirmed to be involved in the induced resistance of maize triggered by *T. harzianum* [3]. The hyd1-induced systemic resistance is mainly related to brassinolactin-mediated signaling, possibly mediated by jasmonic acid/ethylene (JA/ET) signaling [47]. A series of enzymes involved in the *AOS* branch, including allene oxide cyclase 12-oxo-phytodienoic acid reductase (*OPR*) and the production of different jasmonates, is associated with plant resistance [37]. Some products of the *HPL* branch, including many volatile C-6 aldehydes and alcohols, or so-called green leaf volatiles (*GLVs*), are associated with plant disease resistance, and they act as signaling molecules in response to pathogens and pests [48,49,50]. Our study suggests that AOS may be involved in *thph2*-mediated defense responses in maize because *AOS* transcripts were upregulated by the T30:GFP and OE*thph2*:GFP strains but remained low in the untreated and ∆*thph2*:GFP-strain-treated plants. In addition, after the plants were treated with the Thph2 protein, the *AOS* and *HPL* expression levels significantly increased, while *ZmPR1*, *ZmPR5*, and *PAL* were almost not expressed.

Furthermore, our study showed that, in the early stages of root colonization by *Trichoderma* regulated by *thph2*, *AOS* and *HPL* were significantly expressed in the leaves when the roots were treated with the *thph2*-overexpressed *Trichoderma* strain, which indicated that *thph2* contributed to the induced systemic resistance achieved by promoting long-distance signal transduction from the roots to the leaves. The phenomenon has also been confirmed by the work conducted by Chuanjin Yu et al. [47]. In summary, the experimental results suggest that *T. harzianum thph2* induced systemic resistance in maize through an ISR mechanism of the JA/ET-mediated signaling pathway.

## 5. Conclusions

In this study, we found that *thph2* was negatively correlated with the number of spores produced by fungi and positively correlated with the growth rate of mycelia. At the same time, we also found that *thph2* could affect the colonization of *T. harzianum* in plant roots through two stages of interaction: in the early stage, *Trichoderma* inhibited the defense gene expression of maize roots, allowing *Trichoderma* to achieve fast colonization inside the root tissue, and in the second stage, *thph2* shifted its function to elicit maize root defense against pathogen infections. *AOS* and *HPL* were the main defense genes in maize that were sensitive to the *thph2* of the *Trichoderma* action. Therefore, we preliminarily concluded that *thph2* could promote the colonization of maize roots by *T. harzianum* and promote ISR at the same time. In the later colonization period, the expression of the SA pathway marker genes *ZmPR1*, *ZmPR5*, and *PAL* increased. The JA pathway marker genes *AOS* and *HPL* were expressed in maize leaves when the roots were inoculated with different *Trichoderma* mutants. The treatment of maize roots with the Thph2 protein also confirmed that the *thph2* of *T. harzianum* is involved in the systemic defense gene expression from the roots to the leaves in maize, where the *Trichoderma-thph2*-induced and JA-mediated pathways in maize might be dominant against foliar diseases.

## Figures and Tables

**Figure 1 jof-09-01168-f001:**
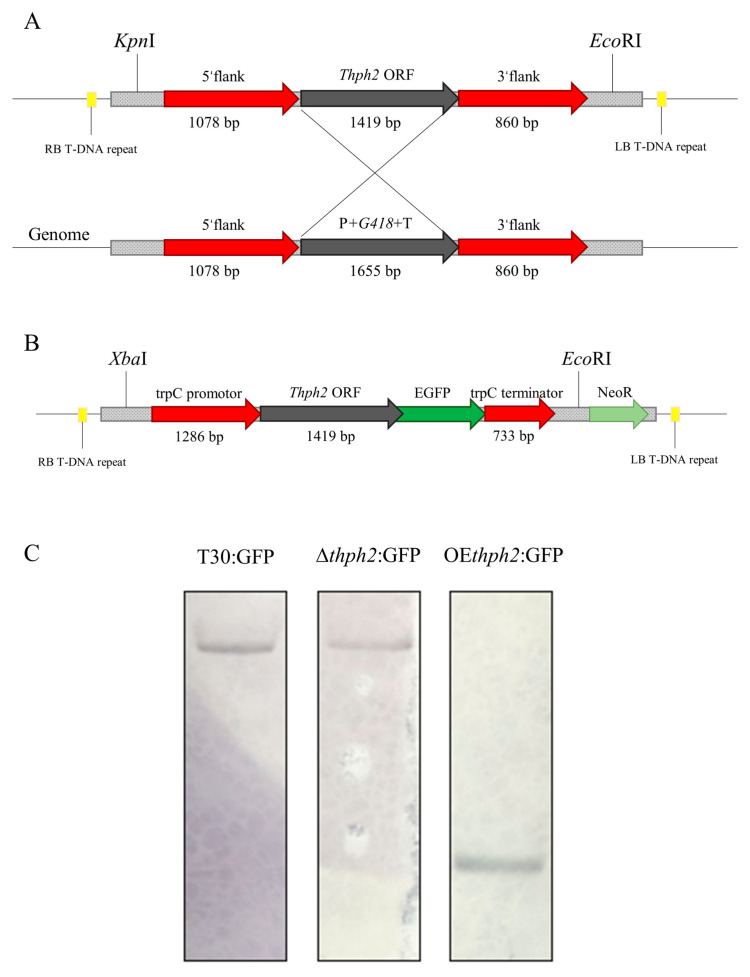
Construction and identification of *thph2*-deletion and -overexpression transformants. (**A**) Schematic diagram of gene-deletion scheme construction. (**B**) Schematic diagram of gene-overexpression scheme construction. (**C**) Southern blotting detected the copy number of G418. All the transformed strains contained the marker gene G418 on the plasmid, which could be used as a probe to verify the copy number of the transformed strains.

**Figure 2 jof-09-01168-f002:**
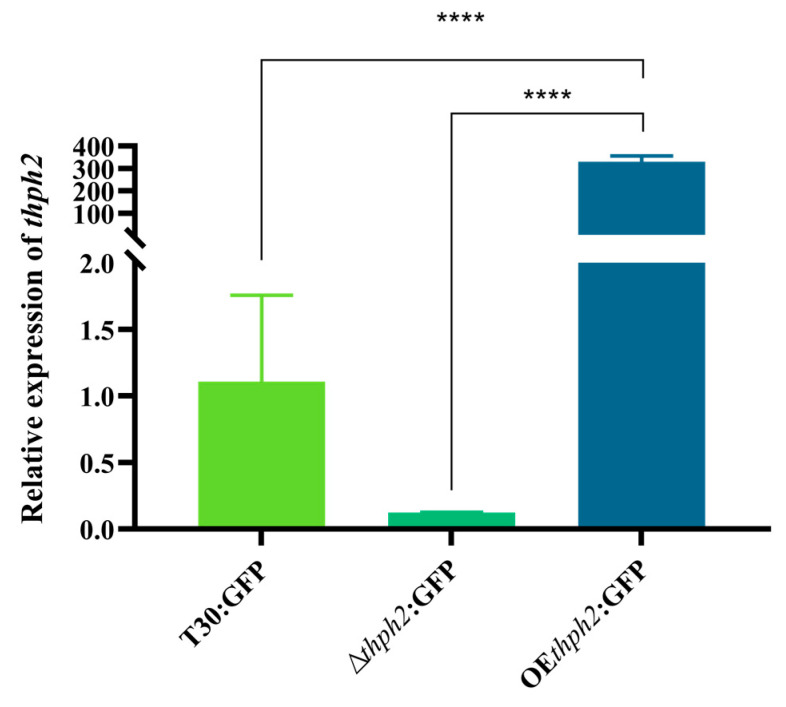
Expression of *thph2* in transformants. The expression of *thph2* was different in different mutants. The data are the means of 5 replicates for each treatment. The asterisk (*) represents a significant difference between the transformants and T30:GFP based on ANOVA (significance: **** *p* ≤ 0.0001).

**Figure 3 jof-09-01168-f003:**
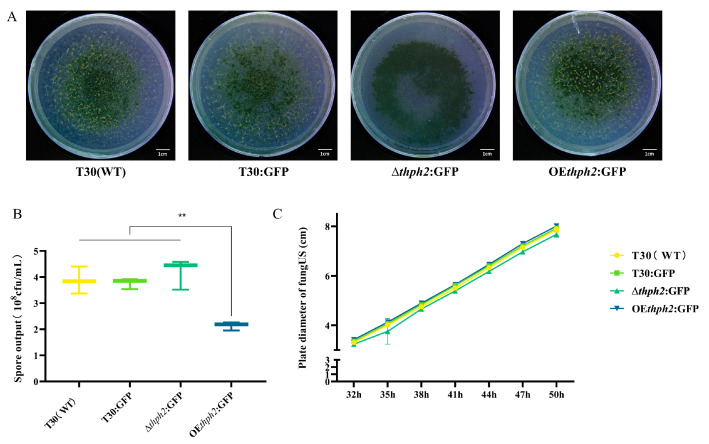
Evaluation of growth, reproduction, and phenotype. (**A**) The growth phenotype difference among different strains. (**B**) The spore output difference among different strains. (**C**) Rate of mycelium growth difference among different strains. The data are the means of 15 replicates for each treatment. The asterisk (*) represents a significant difference between the transformants and T30:GFP based on ANOVA (significance: ** *p* ≤ 0.01).

**Figure 4 jof-09-01168-f004:**
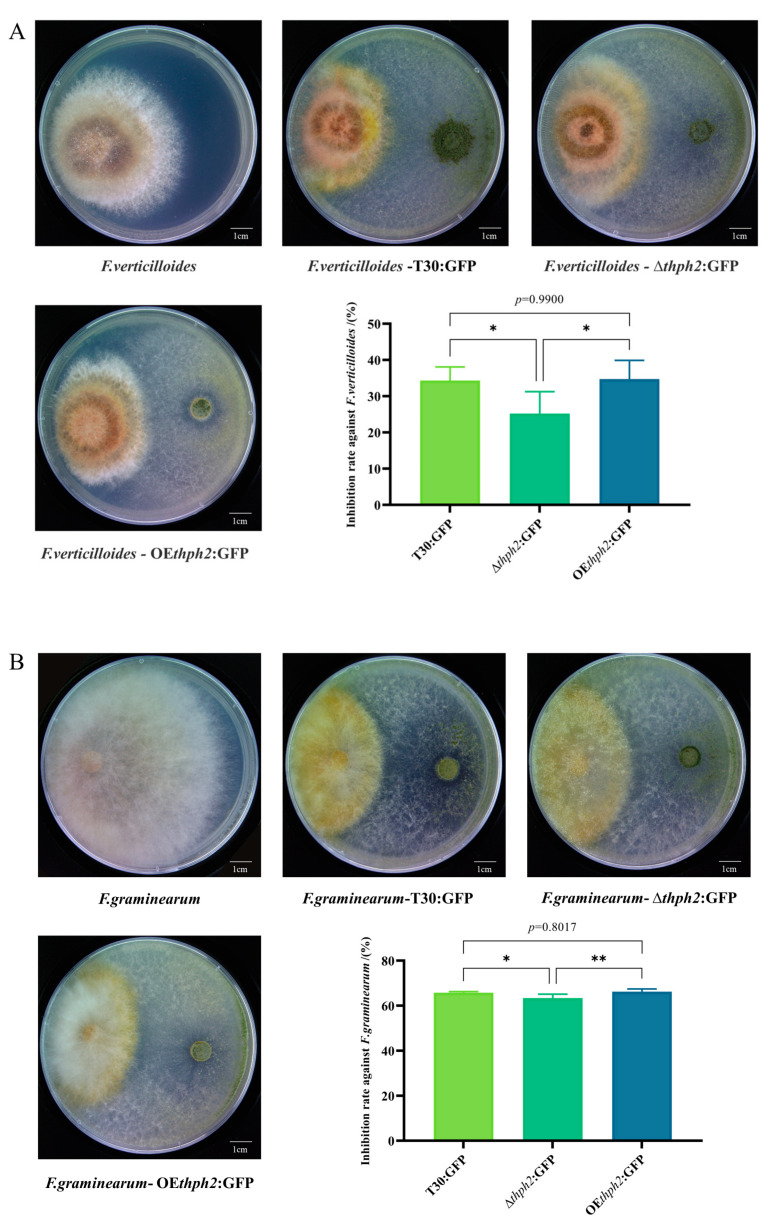
Evaluation of the antagonism of different *Trichederma* mutant strains toward pathogens. (**A**) Antagonistic plate and data analysis of T30:GFP, ∆*thph2*:GFP, and OE*thph2*:GFP against *F. verticilloides*. (**B**) Antagonistic plate and data analysis of T30:GFP, ∆*thph2*:GFP, and OE*thph2*:GFP against *F. graminearum*. (**C**) Antagonistic plate and data analysis of T30:GFP, ∆*thph2*:GFP, and OE*thph2*:GFP against *C. heterostrophus*. The results are the means of 5 replicates for each treatment. The asterisk (*) represents a significant difference between the transformants and T30:GFP (*p* < 0.05) based on ANOVA (significance: * *p* ≤ 0.05, ** *p* ≤ 0.01).

**Figure 5 jof-09-01168-f005:**
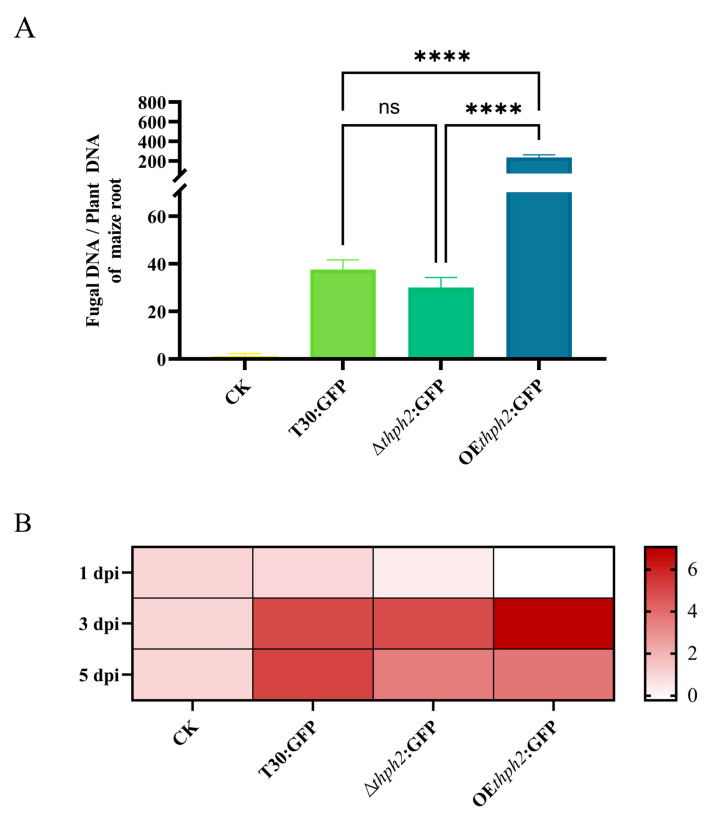
Colonization by T30:GFP, ∆*thph2*:GFP, and OE*thph2*:GFP in the roots of maize plants. (**A**) After 3 days of inoculation, the colonization rate of fungal DNA and plant DNA was determined by qRT-PCR. (**B**) After 1 day, 3 days, and 5 days of inoculation, the heat map of the colonization rate of fungal DNA and plant DNA was determined by qRT-PCR. The results are the means of 6 replicates for each treatment, and 3 biological replicates were performed (significance: **** *p* ≤ 0.0001; ns: no significance).

**Figure 6 jof-09-01168-f006:**
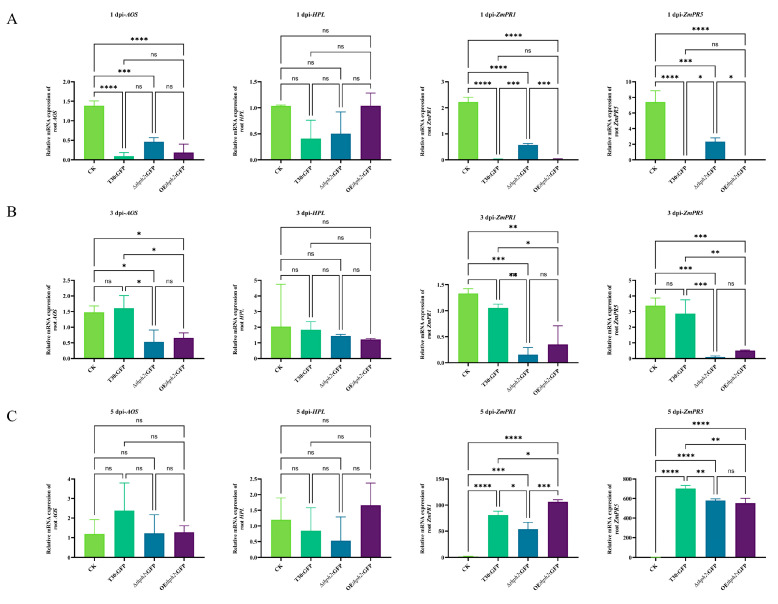
Expression of defense response genes in maize roots after colonization by *Trichoderma*. (**A**) Expression analysis of maize-root-related defense response genes *ZmAOS* and *ZmHPL* (JA) and *ZmPR1* and *ZmPR5* (SA) 1 day after inoculation. (**B**) Expression analysis of maize-root-related defense response genes *ZmAOS* and *ZmHPL* (JA) and *ZmPR1* and *ZmPR5* (SA) 3 days after inoculation. (**C**) Expression analysis of maize-root-related defense response genes *ZmAOS* and *ZmHPL* (JA) and *ZmPR1* and *ZmPR5* (SA) 5 days after inoculation. The results are the means of 6 replicates for each treatment, and 3 biological replicates were performed (significance: * *p* ≤ 0.05, ** *p* ≤ 0.01, *** *p* ≤ 0.001, **** *p* ≤ 0.0001; ns: no significance).

**Figure 7 jof-09-01168-f007:**
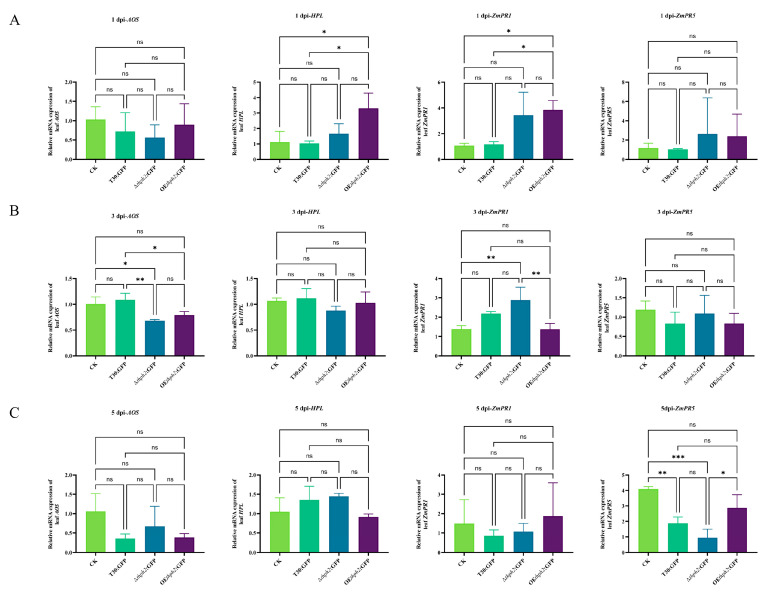
Expression of defense response genes in maize leaves after colonization by *Trichoderma*. (**A**) Expression analysis of maize-root-related defense response genes *ZmAOS* and *ZmHPL* (JA) and *ZmPR1* and *ZmPR5* (SA) 1 day after inoculation. (**B**) Expression analysis of maize-root-related defense response genes *ZmAOS* and *ZmHPL* (JA) and *ZmPR1* and *ZmPR5* (SA) 3 days after inoculation. (**C**) Expression analysis of maize-root-related defense response genes *ZmAOS* and *ZmHPL* (JA) and *ZmPR1* and *ZmPR5* (SA) 5 days after inoculation. The results are the means of 6 replicates for each treatment, and 3 biological replicates were performed (significance: * *p* ≤ 0.05, ** *p* ≤ 0.01, *** *p* ≤ 0.001; ns: no significance).

**Figure 8 jof-09-01168-f008:**
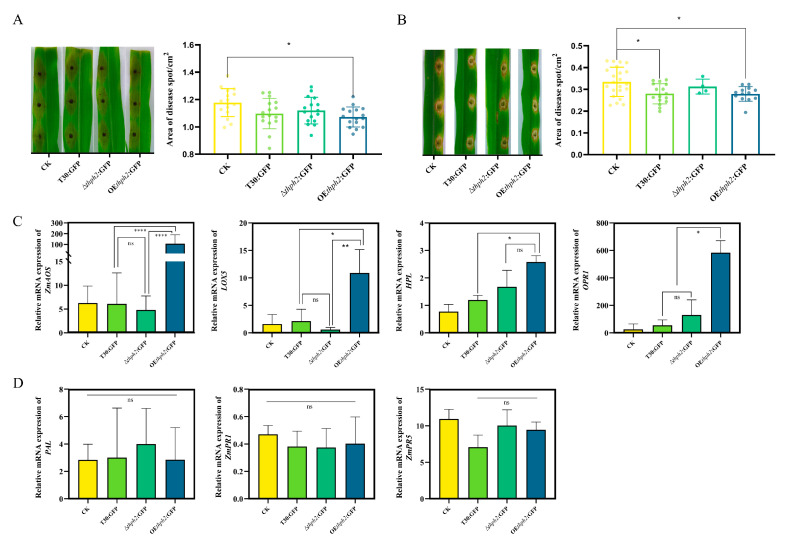
Effect of T30:GFP, ∆*thph2*:GFP, and OE*thph2*:GFP strains on ISR in maize seedlings to *C. heterostrophus* and expression of defense-related genes in the leaves of T30:GFP-, ∆*thph2*:GFP-, and OE*thph2*:GFP-strain-induced maize plants challenged with *C. heterostrophus*. (**A**) Phenotypic map and data analysis of maize leaves inoculated in vitro with different mutant strains. (**B**) Phenotypic map and data analysis of maize leaves inoculated with different mutant strains by a live inoculation. (**C**) *LOX5*, *HPL*, *AOS*, and *OPR1* gene expression at 36 h after the inoculation of *C. heterostrophus*. (**D**) *ZmPR1*, *ZmPR5*, and *PAL* gene expression at 36 h after the inoculation of *C. heterostrophus*. The results are the means of 5 replicates for each treatment; the value is the standard error of the mean. Different letters above the bars indicate significant differences based on ANOVA (significance: * *p* ≤ 0.05, ** *p* ≤ 0.01, **** *p* ≤ 0.0001; ns: no significance).

**Figure 9 jof-09-01168-f009:**
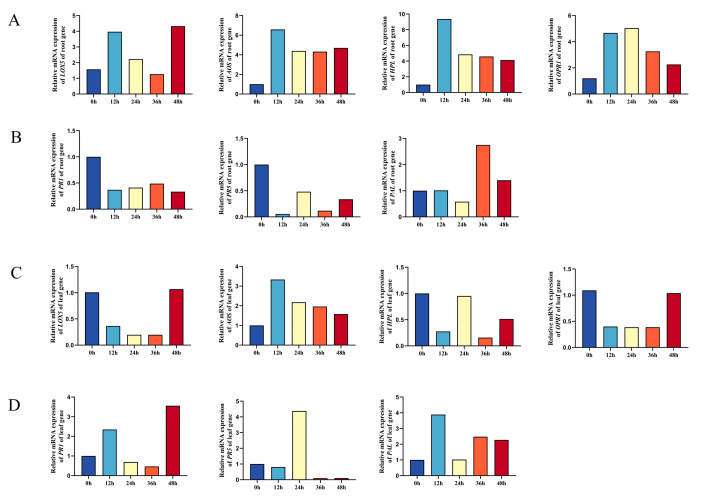
Thph2 protein induced defense response gene expression in maize roots and leaves. (**A**) Thph2 protein induced the expression of the defense response genes *LOX5*, *AOS*, *HPL*, and *OPR1* in maize roots. (**B**) Thph2 protein induced the expression of the defense response genes *ZmPR1*, *ZmPR5*, and *PAL* in maize roots. (**C**) Thph2 protein induced the expression of the defense response genes *LOX5*, *AOS*, *HPL*, and *OPR1* in maize leaves. (**D**) Thph2 protein induced the expression of the defense response genes *ZmPR1*, *ZmPR5*, and *PAL* in maize leaves. The results are the means of 6 replicates for each treatment, and 3 biological replicates were performed.

## Data Availability

Data will be available upon request.

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
