# Peer review of "Trichoderma harzianum Cellulase Gene thph2 Affects Trichoderma Root Colonization and Induces Resistance to Southern Leaf Blight in Maize"

_jof, 2023, doi:10.3390/jof9121168_

Round 1
Reviewer 1 Report (Previous Reviewer 1)
Comments and Suggestions for Authors
Authors carefully revised this manuscript according to reviewer's comments, therefore I recommend to accept this manuscript to Journal of Fungi.
Author Response
Dear editor and dear reviewers:
We appreciate you for spending time to review our paper and providing valuable comments.
Thank you for your review!
Reviewer 2 Report (New Reviewer)
Comments and Suggestions for Authors
Dear authors, below I share observations on your manuscript that can help improve it:
· There are grammatical errors in the manuscript that cause the meaning of the sentences to be lost. A revision of the English is suggested.
· Verify the title of the manuscript is appropriate or represents your work.
· Delete the paragraph from lines 28 to 34.
· Review in the paragraph of lines 68 to 71 what was wanted to be stated.
· In the paragraph from lines 90 to 97 it is not clear what work was done before and what is new in this manuscript.
· I consider that the materials and methods sections must be organized chronologically, as the experiments were carried out.
· Check that the line spacing is the same.
· In Figure 1 in the observed bands, it is not clear why the large differences with the detected molecular sizes. Southern blotting of the three transformants is generally performed in the same assay.
· Figure 1B does not show where the selection marker is in the diagram, which is very confusing. How did you obtain the transformants without a selection marker?
· I consider that the T-DNA scheme of the pCambia-1300:eGFP plasmid must be added.
· On line 285 they mention the destruction of the thph2 gene. What does this mean?
· How was the expression analysis in Figure 2 carried out? Why is there expression of the thph2 gene in the Δthph2:GFP mutant? Under what growth condition was this analysis performed?
· Under what growth conditions was the analysis carried out in Figure 3. As it is shown that the effect of the overexpressor of producing fewer spores is not because a gene involved in spore formation was affected, this with the random integration of the OEthph2:GFP vector.
· The images in figure 4 are very small, they have to be made larger to be able to observe and analyze.
· The colonization of the OEthph2:GFP strain is greater than that of the T30:GFP and Δthph2:GFP strains. How can be, because in the Petri dish growth assays the diameter of the colonies of the strains was the same?
· The quality of the figures in image 6 is very poor, so they cannot be analyzed.
· In figure 7, as well as in figure 6, it is not observed which gene each graph corresponds.
· Improve the images in figure 8.
· The discussion occupies greater clarity and explains the results obtained. It remains to contrast the results obtained from the study of the thph2 gene and its protein against other studies reported on similar proteins in other fungi.
I suggest major corrections to the entire manuscript and its resubmission.
Comments on the Quality of English Language· There are grammatical errors in the manuscript that cause the meaning of the sentences to be lost. A revision of the English is suggested.
Author Response
Dear editor and dear reviewers
Re: Manuscript ID: jof-2430584 and Title: “Trichoderma harzianum cellulase gene thph2 affects Trichoderma root colonization and induces resistance to southern leaf blight in maize”.We appreciate you for spending time to review our paper and providing valuable comments. Your valuable and insightful comments that led to possible improvements in the current version.The authors carefully considered the comments and tried our best efforts to address every one of them. The authors welcome further constructive comments if any. We provided the point-by-point response to reviewer’s concerns. We hope that you find our responses satisfactory and that the manuscript is now acceptable for publication.
Reviewer #2:
1.Commnet:
There are grammatical errors in the manuscript that cause the meaning of the sentences to be lost. A revision of the English is suggested.
Response:
I have made revisions to the language of this manuscript and used a paid editing service at https://www.mdpi.com/authors/english
2.Commnet:
Verify the title of the manuscript is appropriate or represents your work.
Response:
I think the title of the manuscript represents and is appropriate my work.
3.Commnet:
Delete the paragraph from lines 28 to 34.
Response:
I have deleted that part.
4.Commnet:
Review in the paragraph of lines 68 to 71 what was wanted to be stated.
Response:
I have made modifications based on the content of the article.
5.Commnet:
In the paragraph from lines 90 to 97 it is not clear what work was done before and what is new in this manuscript.
Response:
I have made revisions in the Manuscript.
6.Commnet:
I consider that the materials and methods sections must be organized chronologically, as the experiments were carried out.
Response:
The materials and methods sections have been reorganized in the order in which the experiments were carried out.
7.Commnet:
Check that the line spacing is the same.
Response:
I have checked the line spacing and made it consistent
8.Commnet:
In Figure 1 in the observed bands, it is not clear why the large differences with the detected molecular sizes. Southern blotting of the three transformants is generally performed in the same assay.
Response:
The restriction enzyme we used was BamHI.Because the insertion method of ATMT was random, the size of fragments cut by enzyme was not fixed, so the southern blotting of the three transformants showed different sizes.
9.Commnet:
Figure 1B does not show where the selection marker is in the diagram, which is very confusing. How did you obtain the transformants without a selection marker?
Response:
The inserted fragment was followed by the genetic mycin resistance gene NeoR, which could be used to screen the transformants.
10.Commnet:
I consider that the T-DNA scheme of the pCambia-1300:eGFP plasmid must be added.
Response:
The vector construction of pCambia-1300:eGFP and transformation are added in 2.3 and 2.4, respectively.
11.Commnet:
On line 285 they mention the destruction of the thph2 gene. What does this mean?
Response:
It means gene thph2 deletion.It has been revised in the manuscript.
12.Commnet:
How was the expression analysis in Figure 2 carried out? Why is there expression of the thph2 gene in the Δthph2:GFP mutant? Under what growth condition was this analysis performed?
Response:
Quantitative real-time PCR may have a no template control. In this case, a Cq value will be generated, but the result of 2-ΔΔCt is infinitely close to 0, so it can be regarded as a successful knockout.By PCR amplification, no thph2 band was found, which can also be used as an evidence of successful deletion.This analysis was performed after 7 days of culture on PDA medium.
13.Commnet:
Under what growth conditions was the analysis carried out in Figure 3. As it is shown that the effect of the overexpressor of producing fewer spores is not because a gene involved in spore formation was affected, this with the random integration of the OEthph2:GFP vector.
Response:
These transformants were cultured under PDA medium. The points have been revised in the manuscript.
14.Commnet:
The images in figure 4 are very small, they have to be made larger to be able to observe and analyze.
Response:
Figure 4 has been improved.
15.Commnet:
The colonization of the OEthph2:GFP strain is greater than that of the T30:GFP and Δthph2:GFP strains. How can be, because in the Petri dish growth assays the diameter of the colonies of the strains was the same?
Response:
PDA medium was used in the Petri dish growth assays to detect the basic growth indexes of transformants.This gene thph2 is related to the hydrolysis of cellulose, and the root surface of maize contains a lot of cellulose, so different transformants have different effects on the colonization of Trichoderma into maize roots.
16.Commnet:
The quality of the figures in image 6 is very poor, so they cannot be analyzed.
Response:
Figure 6 has been partially revised, and analysis has been added to the manuscript.
17.Commnet:
In figure 7, as well as in figure 6, it is not observed which gene each graph corresponds.
Response:
Figure 7 has been partially revised, and analysis has been added to the manuscript.
18.Commnet:
Improve the images in figure 8.
Response:
Figure 8 has been improved.
19.Commnet:
The discussion occupies greater clarity and explains the results obtained. It remains to contrast the results obtained from the study of the thph2 gene and its protein against other studies reported on similar proteins in other fungi.
Response:
I have carefully considered your comments and added some contents in the discussion section.
20.Commnet:
I suggest major corrections to the entire manuscript and its resubmission.
Response:
I have modified manuscript according to your suggestions, thank you for your comments.

Round 2
Reviewer 2 Report (New Reviewer)
Comments and Suggestions for Authors
Dear authors, I consider that the article has improved with the changes made, I share my new observations:
On line 86 modify ja to uppercase jasmonic acid (JA).
The access numbers to the databases of the T. harzianum genes used, as well as the oligonucleotides used in T. harzianum, remain to be included.
It is striking that the overexpressing protein was fused to the GFP protein as shown in the scheme in Figure 1B. This fusion does not affect the function of the THPH2 protein? What are the domains of the THPH2 protein?
How the fusion of the THP2 protein to GFP was carried out?
Review the paragraph from lines 249 to 258 and delete the text corresponding to the format file.
The images in the figures have very low quality, the text within the images is distorted and pixelated. Improve the quality of your images.
The discussion does not mention the work previously published by your working group, which is very similar. I consider that they have to discuss or mention what is relevant about this manuscript as opposed to previously published work.
Saravanakumar, K. et al. Cellulase from Trichoderma harzianum interacts with roots and triggers induced systemic resistance to foliar disease in maize. Sci. Rep. 6, 35543; doi: 10.1038/srep35543 (2016).
Author Response
November 14, 2023
Dear editor and dear reviewers
Re: Manuscript ID: jof-2430584 and Title: “Trichoderma harzianum cellulase gene thph2 affects Trichoderma root colonization and induces resistance to southern leaf blight in maize”.We appreciate you for spending time to review our paper and providing valuable comments. Your valuable and insightful comments that led to possible improvements in the current version.The authors carefully considered the comments and tried our best efforts to address every one of them. The authors welcome further constructive comments if any. We provided the point-by-point response to reviewer’s concerns. We hope that you find our responses satisfactory and that the manuscript is now acceptable for publication.
Reviewer #2:
1.Commnet:
On line 86 modify ja to uppercase jasmonic acid (JA).
Response:
I have made revisions in the Manuscript.
2.Commnet:
The access numbers to the databases of the T. harzianum genes used, as well as the oligonucleotides used in T. harzianum, remain to be included.
Response:
Trichoderma afroharzianum CGMCC22479 is stored in China General Microbiological Culture Collection Center. I've uploaded the thph2 sequence to NCBI, the submission ID is 2762804 and the accession number is BankIt2762804 Seq1 OR795025. Supplementary information 1 is the sequence of thph2.
- Commnet:
It is striking that the overexpressing protein was fused to the GFP protein as shown in the scheme in Figure 1B. This fusion does not affect the function of the THPH2 protein? What are the domains of the THPH2 protein?
Response:
Thph2 has two main domains, the binding domain and the catalytic domain. The combination of GFP does not break these two domains. These two domains also have binding and catalytic functions, respectively. The detailed structure is in Supplementary information 2.Enzyme activity detection of OEthph2:GFP and protein Thph2(Fig 1).
Figure 1. Enzyme activity detection of OEthph2:GFP and protein Thph2
- Commnet:
How the fusion of the THPH2 protein to GFP was carried out?
Response:
Remove the thph2 stop codon TGA and connect the GFP-ORF directly.
- Commnet:
Review the paragraph from lines 249 to 258 and delete the text corresponding to the format file.
Response:
I have deleted words not related to the manuscript
- Commnet:
The images in the figures have very low quality, the text within the images is distorted and pixelated. Improve the quality of your images.
Response:
I have improved the quality of your images.
- Commnet:
The discussion does not mention the work previously published by your working group, which is very similar. I consider that they have to discuss or mention what is relevant about this manuscript as opposed to previously published work.
Saravanakumar, K. et al. Cellulase from Trichoderma harzianum interacts with roots and triggers induced systemic resistance to foliar disease in maize. Sci. Rep. 6, 35543; doi: 10.1038/srep35543 (2016).
Response:
I have added discussion up to lines 473-478 and lines 489-495.
We have tried our best to revise our manuscript according to the comments.which we would like to submitfor your kind consideration. We would like to express our greatappreciation to you and reviewers for comments on our paperooking forward to hearing from you.Thank you and best regards.
Yours sincerely,Bo Lang
Corresponding author:Jie Chen
E-mail: [email protected]

This manuscript is a resubmission of an earlier submission. The following is a list of the peer review reports and author responses from that submission.
Round 1
Reviewer 1 Report
Comments and Suggestions for Authors
1. First paragraph of introduction should be removed
2. Line 92: use "surface disinfected" in stead of "sterilized"
3. Line 239: author did not describe southern blot in material and method section
4. Line 288: Why percentage inhibition is higher than 100%
5. This study showed over expression of gene how about end product such as enzyme activity of bioassay?
Comments on the Quality of English Language
This manuscript neesd extensive English editing by native.
Reviewer 2 Report
Comments and Suggestions for Authors
See atached file

Comments on the Quality of English LanguageSee atached file